# Alkylresorcinols as New Modulators of the Metabolic Activity of the Gut Microbiota

**DOI:** 10.3390/ijms241814206

**Published:** 2023-09-18

**Authors:** Anastasia A. Zabolotneva, Andrei M. Gaponov, Sergey A. Roumiantsev, Ilya Yu. Vasiliev, Tatiana V. Grigoryeva, Oleg I. Kit, Elena Yu. Zlatnik, Aleksey Yu. Maksimov, Anna S. Goncharova, Inna A. Novikova, Svetlana A. Appolonova, Pavel A. Markin, Aleksandr V. Shestopalov

**Affiliations:** 1Department of Biochemistry and Molecular Biology, Faculty of Medicine, N. I. Pirogov Russian National Research Medical University, 1 Ostrovitianov Str., Moscow 117997, Russia; s_roumiantsev@mail.ru (S.A.R.); al-shest@yandex.ru (A.V.S.); 2Russian National Medical Research Center for Endocrinology, 11 Dm. Ulyanova Str., Moscow 117036, Russia; 3Center for Digital and Translational Biomedicine «Center for Molecular Health», 32 Nakhimovskiy prospekt, Moscow 117218, Russia; 4Institute of Fundamental Medicine and Biology, Kazan Federal University, 18 Kremlyovskaya Street, Kazan 420008, Russia; 5National Medical Research Centre for Oncology, 14 Line, 63, Rostov-on-Don 344019, Russiaelena-zlatnik@mail.ru (E.Y.Z.); maximovalexei60@gmail.com (A.Y.M.); fateyeva_a_s@list.ru (A.S.G.); novikovainna@yahoo.com (I.A.N.); 6Laboratory of Pharmacokinetics and Metabolomic Analysis, Institute of Translational Medicine and Biotechnology, Sechenov First Moscow State Medical University, 2-4 Bolshaya Pirogovskaya St., Moscow 119991, Russia; appolonova_s_a@staff.sechenov.ru (S.A.A.); markin_p_a@staff.sechenov.ru (P.A.M.); 7Rogachev National Medical Research Center of Pediatric Hematology, Oncology and Immunology, 1 Samory Mashela Str., Moscow 117997, Russia

**Keywords:** alkylresorcinols, olivetol, gut microbiota, faecal microbiota transplantation, reconstruction of gut microbiota metabolic activity

## Abstract

Alkylresorcinols (ARs) are polyphenolic compounds with a wide spectrum of biological activities and are potentially involved in the regulation of host metabolism. The present study aims to establish whether ARs can be produced by the human gut microbiota and to evaluate alterations in content in stool samples as well as metabolic activity of the gut microbiota of C57BL, *db/db*, and LDLR (−/−) mice according to diet specifications and olivetol (5-n-pentylresorcinol) supplementation to estimate the regulatory potential of ARs. Gas chromatography with mass spectrometric detection was used to quantitatively analyse AR levels in mouse stool samples; faecal microbiota transplantation (FMT) from human donors to germ-free mice was performed to determine whether the intestinal microbiota could produce AR molecules; metagenome sequencing analysis of the mouse gut microbiota followed by reconstruction of its metabolic activity was performed to investigate olivetol’s regulatory potential. A significant increase in the amounts of individual members of AR homologues in stool samples was revealed 14 days after FMT. Supplementation of 5-n-Pentylresorcinol to a regular diet influences the amounts of several ARs in the stool of C57BL/6 and LDLR (−/−) but not *db/db* mice, and caused a significant change in the predicted metabolic activity of the intestinal microbiota of C57BL/6 and LDLR (−/−) but not *db/db* mice. For the first time, we have shown that several ARs can be produced by the intestinal microbiota. Taking into account the dependence of AR levels in the gut on olivetol supplementation and microbiota metabolic activity, AR can be assumed to be potential *quorum-sensing* molecules, which also influence gut microbiota composition and host metabolism.

## 1. Introduction

In recent years, we have observed an increasing interest in ‘host–gut microbiota’ relationship research. The microbiota is a collection of various microorganisms (mainly bacteria) that normally reside in host tissues [1]. The major microbiota community inhabits the gastrointestinal tract, where it plays a barrier role, as well as acts as a regulator of host immunity and metabolism, which is confirmed by the strong correlations between intestinal dysbiosis and certain inflammatory and metabolic diseases, such as obesity, diabetes mellitus, inflammatory bowel disease, etc. [2]. A wide variety of bacterial metabolites are involved in host–microbiota interactions through their signalling and regulatory capabilities. Alkylresorcinols (ARs) are highly lipophilic polyphenols synthesised by bacteria, fungi, some animals, and higher plants [2] through the activity of type III polyketide synthase (PKS) [3].

Humans receive AR primarily from grain meal, as well as components of food preservatives, drugs, or cosmetics [4]. However, it should be considered that ARs can potentially be synthesised by many bacteria, including those that constitute the human gut microbiota, although to our knowledge there is no strict evidence of the bacterial origin of ARs in humans. According to the results of the investigation of the ‘sequence–function relationship’, 45% of the Bacteria phyla possess at least one type III PKS gene [3]. In the study of Funabashi M. [5], Srs-like operon structures coding for type III PKS were found among many Gram-positive and Gram-negative bacteria, indicating that Ars production is a common feature of different prokaryotes. Experimentally, it has been characterised in seventeen bacterial type III PKSs from Actinobacteria, Proteobacteria, Firmicutes, and Cyanobacteria [3]. In bacteria, phenolic lipids serve as precursors for antibiotics, substances that confer antibiotic resistance, or regulators of antibiotic production [6], as well as UV-protective pigments [7] and alternative electron carriers [8]. However, entering the human body Ars possess many regulatory functions including anticancer, anti-inflammatory, antimicrobial, antiparasitic, antioxidant, and genotoxic activities [2,9,10]. In this context, alkylresorcinols (Ars) can be considered as a connecting link between diet, gut microbiome, and host metabolism, as well as potential *quorum sensing* (QS) molecules.

Furthermore, according to published data, most dietary polyphenols are transformed into the colon by the intestinal microbiota before absorption [11]. This conversion is often essential for the absorption of nutrients and modulates the biological activity of these dietary compounds [12]. Gut bacteria can hydrolyse glycosides, glucuronides, sulphates, amides, esters, and lactones [11]. They also perform rings cleavage, reduction, decarboxylation, demethylation, and dehydroxylation reactions [13,14]. Selma et al. perfectly reviewed the variety of species of the gut microbiome, including *Bifidobacterium* sp., *Lactobacillus* sp., *Clostridium* sp., *Eubacterium* sp., etc., and observed their ability to metabolise different polyphenolic structures, converting them into low-absorbable and biologically active compounds [15]. It seems possible that Ars of complex structure ingested with food or originating in the gut might be converted to metabolites with a simpler structure, which are then absorbed into the host’s circulation and influence its metabolism. On the other hand, phenolic compounds also have antimicrobial properties and can interact with the gut microbiota, thus modulating the microbial population of the gastrointestinal tract (GI), affecting both the health of the GI tract and the metabolism of dietary phenolic compounds. For example, in the study by Nikolaev et al., AR (in particular 4-hexylresorcinol) in combination with antibiotics was shown to significantly decrease the number of germinating spores of *Bacillus cereus* in liquid medium, as well as agar medium, in contrast to treatment with antibiotics alone [16]. This observation coincides with other studies showing the potential of Ars as autoregulators [17]. As reported by Oishi et al. [18], Ars significantly increased the amount of *Prevotella* and reduced the amount of *Enterococcus* in the faecal microbiota of mice. Additionally, some polyphenols from beverages, fruits, and vegetables are known to promote or inhibit intestinal bacterial growth [19]. 

The regulatory activity of Ars can be mediated by their binding to the endocannabinoid receptors CB1 and CB2 [20]. CB1 receptors are expressed primarily in the central nervous system, neuronal tissues, and gastrointestinal (GI) tract, while CB2 receptors are present predominantly in peripheral immune cells, such as B lymphocytes, macrophages, mast cells, natural killer cells, lymphatic organs, spleen, tonsils, and thymus [21]. It should be noted that almost all tissues can express endocannabinoid receptors; therefore, the endocannabinoid signalling system (or endocannabinoidome) is a critical homeostasis support system. The endocannabinoid (eCB) system of the GI tract is of particular importance in the context of the ‘host–microbiota–environment’ interrelationship. The extensive study of functions of the eCB system has led to the discovery of several lipophilic molecules—ligands of CB receptors with different and often antagonistic properties [20,22]. It has become clear that eCB plays a principal role in the maintenance of gut homeostasis and its barrier function [23], and it is also implicated in the modulation of gut microbiota composition [24,25]. Olivetol (5-n-pentylresorcinol) is a resorcinol with a short alkyl chain and is one of the end products of type III PKS that is responsible for the synthesis of cannabinoids in plants of the *Cannabis* genus [26]. Olivetol and its derivatives are known to exhibit the properties of antagonists (or competitive inhibitors) of the CB1 and CB2 receptors [20,27,28]. In recent years, many intriguing effects of olivetol have been revealed, such as anticancer activity in breast adenocarcinoma cells [29] and obesoprotective capabilities in diet-induced obesity in mice [30]. The authors claim that the effects of 5-pentylresorcinol are associated with specifically activated PGC-1α deacetylation, increased mitochondrial numbers, and increased fat burning through the induction of olivetol-related lipid metabolism genes [30].

In order to study the potential of Ars as signalling and regulatory molecules, we hypothesised the existence of synthesis of Ars in the human gut by the microbiota. Thus, the objectives of our study were (1) to show the possibility of synthesis of Ars by bacteria in the human gut and (2) to estimate the influence of olivetol supplementation on the metabolic activity of the mouse gut microbiota and AR profiles in mouse faeces depending on diet or genetic background. 

## 2. Results

### 2.1. Estimation of AR Content in Germ-Free Mouse Faeces after FMT

To investigate whether Ars have gut microbiota origin or come from a diet, we performed faecal microbiota transplantation (FMT) of three adult human donors in germ-free mice of the Balb/c lineage. A significant increase in the amounts of individual members of the homologous series of Ars in stool samples was revealed in 14 days after FMT (Figure 1).

Taking into account that the animals were on the same diet and consumed almost the same amount of food, and that mouse cells cannot synthesise any Ars molecules, a significant increase in the content of C3 (Figure 1a), C12 (Figure 1b), and C15 (Figure 1c) (*p* ≤ 0.01) in the faeces indicates a high probability of synthesis of these Ars by representatives of the human gut microbiota transplanted into gnotobiotic mice.

### 2.2. Estimation of AR Content in C57BL/6 and Db/Db and Ldlr (−/−) Mice’s Faeces Depending on Diet Content

The fact that AR can be synthesised in the intestine led us to investigate the regulatory properties of one of the short alkyl chain-length resorcinol—olivetol (C5), which is known to possess inhibitory effects on some enzymes [29] and receptors [20], as well as acting as a regulator of gene transcription [30]. Mice of C57BL/6 line were fed a regular chow diet (RCD) with or without C5 supplementation and a high-fat diet (HFD) with or without C5 supplementation, whereas LDLR (−/−) and *db/db* mice (that were used for the modelling of atherosclerosis and obesity) were fed RCD with or without C5 supplementation only (Figure 2). 

We detected significant differences (*p* < 0.05) in the AR profile in stool samples from C57BL/6 mice that received RCD in dependence on C5 supplementation (Figure 3).

The amounts of C1, C2, C3, C12, and C15 in stool samples of mice nourished with C5 were significantly lower than those from C5-nourished mice, while in contrast the amount of C0 increased after C5 supplementation (Figure 3a–f). 

We have also compared stool Ars profiles of C57BL/6 mice nourished by HFD and HFD with C5. Surprisingly, we have not detected significant differences in AR content in stool samples after C5 supplementation (Figure 3a–f). However, there were differences (*p* < 0.05) in amounts of C0 and C15 between the RCD and HFD nourished groups: C0 increased in the HFD group (Figure 3a), while C15 decreased in the HFD compared to the RCD group (Figure 3f). 

Analysis of stool samples from db/db mice has not revealed any significant differences between AR content in stool after RCD or RCD + C5 nutrition (Figure 4a). However, LDLR (−/−) mice showed significant differences in the content of C6, C12 and C15 (Figure 4b).

Again, the amounts of C6, C12, and C15 in the stool samples of mice decreased after feeding RCD with olivetol supplement (Figure 5a–c).

### 2.3. Prediction of Gut Microbiota Metabolic Activity

To estimate the influence of C5 supplementation on mouse gut microbiota metabolic activity, we performed a high-throughput sequencing analysis of the V3–V4 region of bacterial 16S rRNA genes for all studied groups of mice (N = 78, sequencing analysis was performed for each stool sample in all groups except for BL/RCD + C5 and DB/RCD + C5, where only nine samples were sequenced in each group; see Appendix A) followed by sequencing data analysis using PICRUSt2 v2.5.2 software that allows prediction of microbiota metabolic activity. Among the four-hundred-twenty-three pathways analysed, only three pathways differed significantly in C57BL/6 mice fed HFD + C5 compared to the HFD group (Figure 6b, Appendix A), while in the RCD + C5 group of C57BL/6 mice, sixty-nine pathways were changed (Figure 6a, Appendix A). In particular, the abundance of only the L-histidine degradation pathway I has increased after C5 supplementation, and the abundance of all other pathways has decreased. Among these decreased pathways, the majority are responsible for anabolic processes, namely amino acid, vitamin, and nucleotide biosynthesis. Again, in *db/db* mice we have not seen any differences in the abundance of metabolic pathways except for two pathways (haem biosynthesis II and mannan degradation), whose abundance decreased in the C5-awarded group (Figure 6c). These results are in agreement with the analysis of the faecal AR profile, showing no differences in AR content in stool samples from C57BL/6 mice fed the HFD + C5 diet and *db/db* mice fed the RCD + C5 diet compared to groups with a diet without C5 supplementation. 

On the contrary, PICRUSt of the sequencing data received from the microbiota analysis of LDLR (−/−) mice revealed significant differences in the abundance of 194 metabolic pathways’ abundance (Figure 7, Appendix A). Here, we have also seen depletion in the representation of biosynthetic pathways, whereas the representation of the pathways of sulphur oxidation and factor 420 biosynthesis increased. 

These observations may indicate a strong dependence of olivetol effects on diet type and microbial metabolic environment, which, in turn, is associated with intestinal microbiota composition. As was shown in different investigations, db/db mice possess an abnormal gut microbiota composition [31,32], and LDLR null mice develop a specific phenotype depending on conditions of sterility or colonisation by the microbiota in the intestine [33,34]. Supplementing RCD with olivetol has led to significant changes in AR profiles and metabolic activity of the intestinal microbiota in C57BL/6 and LDLR (−/−) mice and had almost no effect on C57BL/6 mice fed a diet of *db/db* or high fat. Therefore, it can be concluded that olivetol may serve as a modulator of intestinal microbiota metabolic activity; however, its effects are determined by the intestinal microbial background and possibly host metabolic characteristics. Furthermore, olivetol serves as a negative regulator of C3, C12, and C15, which we have shown may be the origin of the gut microbiota. 

## 3. Discussion

During recent decades, interest in studying phenolic lipids has increased significantly. Polyphenols, including ARs, constitute a large group of bioactive molecules that are known to confer health benefits, as well as may have an impact on the composition and functioning of the gut microbiome [35]. Several homologues of the AR family have been shown to possess anticancerogenic, obesoprotective, anti-inflammatory, antioxidant, and some other effects [9,10,35,36]. For example, 5-n-pentylresorcinol may be involved in the induction of genes for lipid metabolism, thus accelerating fat burning and protecting against diet-induced obesity [30]. However, ARs as metabolites of bacteria may be synthesised in the human gut, thus influencing the metabolic activity of the gut microbiota, and acting as signalling molecules for the host organism as well. 

For the first time, we have shown the possibility of AR synthesis in the human gut using FMT from human donors to germ-free mice. According to our investigation, the human microbiota can produce 5-n-propyl, 5-n-dodecyl, and 5-n-pentadecylresorcinol among the examined ones (Figure 1). In particular, these types of AR were found to positively correlate with LPS concentrations in blood serum in humans, as shown in a previous study [37]. 

Considering that C3, C12, and C15 could have a bacterial origin, it is especially important to understand why their amounts decreased after C5 supplementation. 

We can provide several explanations for the observed results. On the one hand, high concentrations of C5 in the intestine can lead to an increase in the solubility of other lipophilic compounds, thus improving their absorption in enterocytes. At the same time, an increase in the amount of C0 may be associated with AR metabolism. Potentially, resorcinol C0 can accumulate during AR degradation under bacterial enzyme activity. Bacteria are known to have a wide spectrum of enzymes to metabolise endogenic or exogenic compounds, including polyphenols and other phenolic lipids, making them more soluble or increasing their biological activity [38,39].

Based on the results of our study, we consider olivetol to influence the composition of the intestinal microbiota and the ability of the microbiota to synthesise AR molecules. 

We have demonstrated the absence of the influence of C5 supplementation on AR content in stool samples obtained from *db/db* mice and C57BL/6 mice nourished by HFD and the decrease in C1, C2, C3, C12, and C15 levels in stool samples from C57BL/6 mice nourished by RCD, as well as the decrease in C6, C12 and C15 levels in stool samples obtained from LDLR (−/−) mice after C5 supplementation. 

ARs are known to have beneficial effects on health. For instance, a high-fat high-sugar diet in mice causes obesity, which wheat AR prevents along with its associated metabolic symptoms [18]. In mice, diet-induced obesity is reduced by ARs because they reduce intestinal cholesterol absorption. However, ARs increase feed efficiency by reducing dietary lipid absorption [40]. Therefore, ARs are involved in lipid metabolism and could have protective properties in obesity. In this investigation, our objective was to study the effects of AR on intestinal microbiota function in relation to diet-induced or genetically determined obesity. *Db/db* mice are known to develop an obese phenotype independently of diet type, while LDLR (−/−) mice do not develop an obese phenotype under the conditions of a regular diet but have abnormal lipid metabolism that leads to the manifestation of atherosclerosis. Different studies have confirmed that obese and lean mice have a different composition and metabolic activity of the gut microbiota [41]. We proposed that microbiota metabolic activity is associated not only with the phenotype, but also with the mouse genotype. Therefore, AR could influence the metabolic activity of the gut microbiota differently in genetically determined or diet-induced obesity. As we can see, the metabolic profile of the mouse microbiota differs significantly after olivetol supplementation. In particular, C5 supplementation inverses the abundance of several pathways that were different from those of mice fed an HFD compared to mice fed an RCD (see Table 1, Appendix A). 

For example, the abundance of the fatty acid synthesis and mannan degradation pathways increased in mice fed a high-fat diet compared to mice fed a regular chow diet, while the abundance of the reductive acetyl coenzyme A pathway and the haem biosynthesis pathway decreased. Supplementation of C5 to an HFD increases the abundance of haem biosynthesis and reductive acetyl coenzyme A pathways, supplementation of C5 to *db/db* mice (who develop obesity as mice fed an HFD do) decreases the mannan degradation pathway, supplementation of C5 to LDLR (−/−) mice (who do not develop obesity) increases the pathways of methanogenesis (superpathway of sulphur oxidation and pathway of factor 420 biosynthesis), which were decreased in mice fed an HFD. These may indicate the ability of C5 to influence the metabolic activity of the microbiota. We can suggest that ARs because of their ability to decrease dietary lipid absorption change the representation of the substate to intestinal microbes, thus promoting bacteria to adapt to a new “fodder base”. However, depending on the specific genetic background, such adaptations may have opposite directions. As we can see in *db/db* and LDLR (−/−) mice, C5 supplementation decreases the abundance of pathways involved in methanogenesis in *db/bd* mice and increases in LDLR (−/−) mice. 

Currently, our knowledge regarding the exact mechanisms of AR action is limited. However, we see great regulatory potential for AR in modulating the composition of the gut microbiota and/or its metabolic activity. Without a doubt, this area of investigation requires meticulous research.

## 4. Materials and Methods

### 4.1. Experimental Animals and Study Design

C57BL/6SPF mice (n = 40, males) (were bred at the Nursery of Laboratory Animals in Puschino, Puschino, Russia), LDLR (−/−) mice (n = 20, males), and db/db (n = 20, males) mice (were bred at JAX-East and JAX-West Nurseries of Laboratory Animals, Sacramento, CA, USA) were acclimated to housing conditions (22 °C, 55% humidity, 12 h:12 h light: dark cycle) in SPF level animal centre of I.M. Sechenov First Moscow State Medical University (Moscow, Russia) with ad libitum access to sterile food (Altromin 1324 FORTI, Lage, Germany) and water for 1 week prior to formal study. After the adaptation period, the mice were divided into groups, each containing 10 individual animals according to genotype, with randomisation according to body weight (a spread in the group by weight of not more than ±10%). At the beginning of the investigation, the age of the mice was 8 weeks, and mean body weight was 19 ± 2 g. The alimentary obesity model was reproduced by feeding laboratory animals a high-fat diet enriched with animal-derived triglycerides and providing up to 30% of total calories (Altromin C 1090-30, Lage, Germany; the food includes: 13.3% crude fat, 21.1% crude protein, 5.1% crude fibre, 3.9% crude ash, 50.8% nitrogen-free extractives and 5.8% moisture) starting from the age of 8 weeks until the end of the experiment for 90 days. The animals in the control group (C57BL/6SPF), as well as the LDLR (−/−) and db/db mice, were fed a regular chow diet (Altromin 1324 FORTI, Lage, Germany) for the entire investigation period. At the beginning of the study, the age of the mice was 8 weeks, and mean body weight was 19 ± 2g. Administration of 5-n-pentylresorcinol (C5) (Hangzhou ROYAL Import & Export Co., Ltd., Hangzhou, Zhejiang province, China) was carried out through an atraumatic intragastric tube at a dose of 2 mg/day per mouse for 90 days. The control groups of the animals received a placebo (1% water solution of pharmaceutical starch) in the same manner. The mice were euthanised after 90 days of feeding after being anesthetised with isofurane (RWD Life Science, Chenzhen, Guangdong, China). The colon tissues were taken sterile. After being quickly frozen in liquid nitrogen, tissue samples were kept at 80 °C until analysis. The colon samples were cut into 1-cm sections under sterile conditions, placed in separate sterile Eppendorf tubes, kept in dry ice, and then sent for high-throughput sequencing analysis (10 samples for each group except for BL/RCD + C5 and DB/RCD + C5 mice where only nine samples were sequenced in each group).

All animal experiments were approved by the Ethics Committee for Animal Research, I.M. Sechenov First Moscow State Medical University, Russia (protocol number 96 from 2 September 2021).

### 4.2. Faecal Microbiota Transplantation

To explore whether the intestinal microbiota could produce AR molecules, we colonised germ-free Balb/c mice (8 to 10 weeks old) obtained from «Taconic Biosciences», New York, USA (n = 40) with the faecal microbiota of adult human donors (n = 3) (see Table 2). 

Mice of both sexes were randomly selected for bacterial colonisation; each group was evenly balanced for male/female ratio. Mice were gavaged with 100 µL of faecal microbiota suspension obtained from human donors three times a day (100 µL/day). To prepare faecal microbiota samples for transplantation, a portion of frozen faeces from a donor was homogenised in a ratio of 0.1 g of stool to 1000 μL of saline solution and then filtered through a paper filter. The filtrate was collected for subsequent administration to mice. Mice in the control group were gavaged with 100 μL of 0.9% NaCl solution three times a day (100 µL/day). A period of acclimatisation of the animals was at least 4 days. The mice were housed for 14 days after transplantation in sterilised ventilated cages in the SPF area of the testing laboratory centre vivarium on a 12 h light/12 h dark cycle with free access to food and water at a temperature of 20 to 23 °C and a humidity of 35 to 75%, with a purified air circulation of 10 to 15 L/h, in groups of 5 animals per cage. All mice were given *ad libitum* access to sterile food (Altromin 1324 FORTI, Germany) and water for the acclimatisation and experiment period of the hole. Mice were anaesthetised with isofurane (RWD Life Science, China) and euthanised within 14 days after microbiota transplantation. The colon tissues were taken sterile. After being quickly frozen in liquid nitrogen, tissue samples were kept at 80 °C until analysis. For GC-MS analysis colon contents were taken. 

Before the start of the experiment, a clinical examination and weighting of the animals were carried out. All experiments were approved by the Ethics Committee for Animal Research of I.M. Sechenov First Moscow State Medical University, Russia (protocol number 96 from 2 September 2021).

All human donors included in the study have signed an informed consent to participate in the survey. The criteria for inclusion of human donors in the study were the absence of antibiotic, prebiotic, and probiotic drug uptake for 3 months prior to the study. Exclusion criteria were severe somatic diseases, any disease of the gastrointestinal tract, any acute conditions, chronic or acute inflammatory diseases, depression, alcoholism, smoking, pregnancy, and the lactation period. The study was carried out in accordance with the Declaration of Helsinki and was approved by the Ethics Council at the National Medical Research (protocol code No 44, date of approval 20 December 2019). For each of the individuals included in the study, a clinical examination, an anthropometric assessment, and a questionnaire survey were performed, and stool samples were taken for laboratory research. Within 15 min of excretion, the stools were freshly collected from each participant and immediately frozen before being stored in liquid nitrogen.

### 4.3. Quantitative Analysis of ARs

Gas chromatography with mass spectrometric detection (GC-MS) was used to perform quantitative analysis of AR levels in stool samples. An artificial matrix containing sodium chloride and bovine serum albumin was used to perform a calibration to determine the quantitative composition. To normalise the derivatisation processes and smooth out the errors in determining the concentrations of the test substances, the calibration was based on the response of the internal standard (4-(benzyloxy)-phenol) introduced into the test samples. The study was carried out using an Agilent 6890 gas chromatograph with an automatic sample introduction system connected to an Agilent 5850 mass spectrometric detector with electron impact ionisation (Agilent Inc., Santa Clara, CA, USA). The Restek Rtx 5 Sil-MS column (Restek Corporation, Bellefonte, PA, USA) was used for chromatography with the following column parameters: 30 m in length, 250 mm in cross-sectional diameter, and 0.25 mm in particle size. The following chromatography parameters were used: sample injection with a flow division of 5:1, helium as the carrier gas, constant flow as the gas supply mode, 1 mL/min flow rate, and 290 °C as the inlet temperature. The thermostat was set at an initial temperature of 80 °C with a hold period of 1 min, before increasing to 320 °C at a rate of 25 °C/min with a hold period of 4 min. The stool samples were lyophilised and then 300 μL of diethyl ether and the internal standard (4-(benzyloxy)-phenol) were added. The mixture was then stirred on a shaker and centrifuged at 2000 rpm. Two sequential liquid–liquid extractions were performed. The organic phase was transferred to disposable tubes and evaporated in a stream of nitrogen. MSTFA, a silylating agent, was then used to derivatise the sample for 30 min at 60 °C. Following the derivatisation process, GC-MS was used to analyse the samples. The selectivity, linearity, accuracy, reproducibility, matrix effect, and analyte stability of the method were validated. Validation was carried out according to the FDA Bioanalytical Method Validation Guidelines. Faeces samples were collected in ethanol and stored in the Biobank at −80 °C until testing.

### 4.4. High-Throughput Sequencing Analysis and Reconstruction of Gut Microbiota Metabolic Activity

The microbiota analysis was carried out by the scientific research laboratory «Multiomics technologies of living systems» (Kazan, Russia). Using the FastDNATM Spin Kit for Faeces (MP Biomedicals, Santa Ana, CA, USA), genomic DNA was extracted from mouse stool samples. The V3-V4 region of the bacterial 16S rRNA gene was amplified using particular primers (see Appendix A). Following AMPure XP Beads-based PCR product purification (Beckman Coulter, Brea, CA, USA, CB55766755), each sample was barcoded using index primers during secondary round PCR amplification. The amplicons concentration was measured using the Qubit dsDNA High Sensivity Assay Kit (Invitrogen, Carlsbad, CA, USA) with a Qubit 2.0 Fluorometer (Invitrogen, Carlsbad, CA, USA). Before sequencing, the samples were combined in an equal mole ratio to finish preparing the libraries.

The libraries were then high throughput sequenced (2 × 300 bp reads) (Illumina Miseq, Illumina, CA, USA). Raw reads were processed using QIIME2 v2023.7.0 [42] and PICRUSt2 v2.5.2 softwares https://huttenhower.sph.harvard.edu/picrust/ (accessed on 12 September 2023). According to the results of sequencing data processing using PICRUSt2 v2.5.2 software, the microbial metabolic pathways encoded by detected bacterial genomes were scored and the most abundant pathways were detected by multiple *t*-test analysis.

### 4.5. Statistical Data Analysis

Statistical processing of the data was carried out using the method of nonparametric statistics using the GraphPad Prism 10 v10.0.2 (171) statistical software package. The mean and standard deviation were used to present all data. All in vivo experimental data were analysed using Welch’s one-way analysis of variance (ANOVA) or multiple Mann–Whitney tests using the two-stage step-up method (Benjamini, Krieger, and Yekutieli) (false discovery rate Q = 5%). *p* values less than 0.05 were considered statistically significant (* *p* < 0.05, ** *p* < 0.01, *** *p* < 0.001). A correlation analysis was performed according to Spearman with an assessment of the statistical significance of the correlation coefficient.

## 5. Conclusions

For the first time, we have shown that 5-n-propylresorcinol, 5-n-dodecylresorcinol, and 5-n-pentadecylresorcinol can be potentially produced by the intestinal microbiota in the human gut. We have also investigated the influence of 5-n-pentylresorcinol dietary supplementation on AR profiles in C57BL/6, LDLR (−/−), and *db/db* mice. According to our study, olivetol supplementation led to a change in AR content in stool samples of C57BL/6 and LDLR (−/−) mice, but not of *db/db* mice. We have also demonstrated a significant influence of olivetol supplementation on microbiota metabolic activity of RCD nourished C57BL/6 and LDLR (−/−) mice, but not HFD nourished C57BL/6 and *db/db* mice. Olivetol primarily caused a decrease in the abundance of anabolic pathways involved in the biosynthesis of amino acids, vitamins, and nucleotides, as well as a decrease in the abundance of some catabolic pathways. However, our study has some limitations. According to the design of the experiment, we kept mice on an RCD for 14 days after faecal microbiota transplantation and analysed the faecal compounds and the gut microbiota received directly from the colon of each of the 10 animals in each group. However, a time-course representation that demonstrates the progressive increase in ARs across all sets of mice is required. Additional research showing direct synthesis of ARs by gut bacteria is needed. The modulatory potential of ARs in the gut microbiota could be studied in further investigations showing exact alterations in the composition of the gut microbiota associated with AR synthesis and identification of species capable of AR synthesis. 

## Figures and Tables

**Figure 1 ijms-24-14206-f001:**
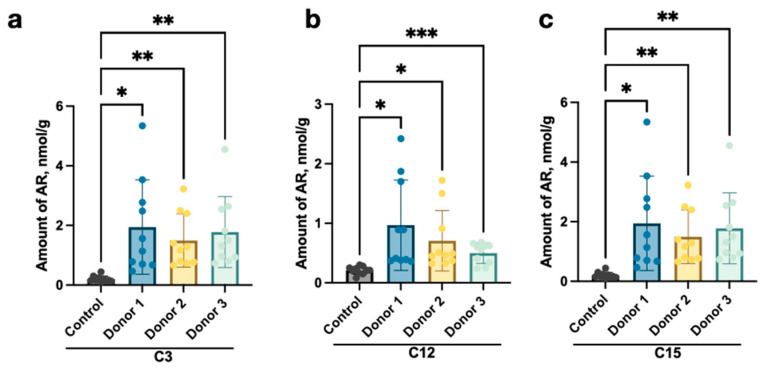
Amounts (nmol/g) of (**a**) propylresorcinol (C3), (**b**) dodecylresorcinol (C12), and (**c**) pentadecylresorcinol (C15) in stool samples of mice after transplantation of the faecal microbiota of human donors (Donor 1–Donor 3) compared to the control group according to Welch’s ANOVA test (* *p* < 0.05, ** *p* < 0.01, *** *p* < 0.001).

**Figure 2 ijms-24-14206-f002:**
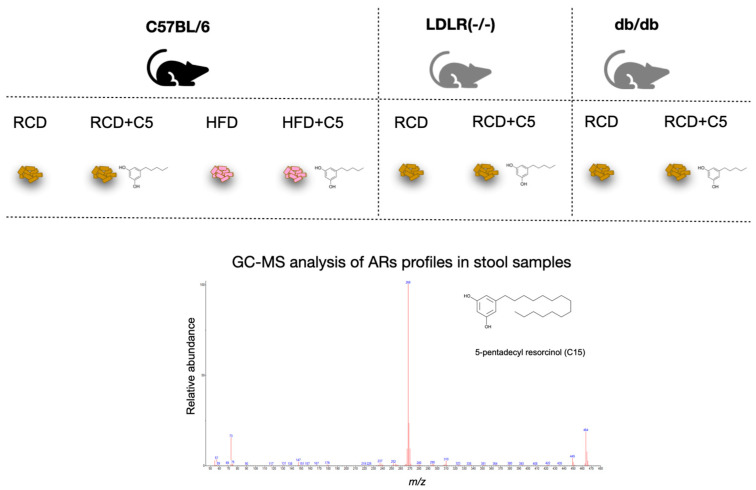
Mice of three lines (C57BL/6, LDLR (−/−), *db/db*) were fed with a regular chow diet (RCD) or a high fat diet (HFD) with (+C5) or without olivetol supplementation. Stool samples from all mice were collected for the following GC-MS analysis of the AR content. The mass spectrum of pentadecylresorcinol is shown as an example. The mass spectra of all Ars are available in Appendix A.

**Figure 3 ijms-24-14206-f003:**
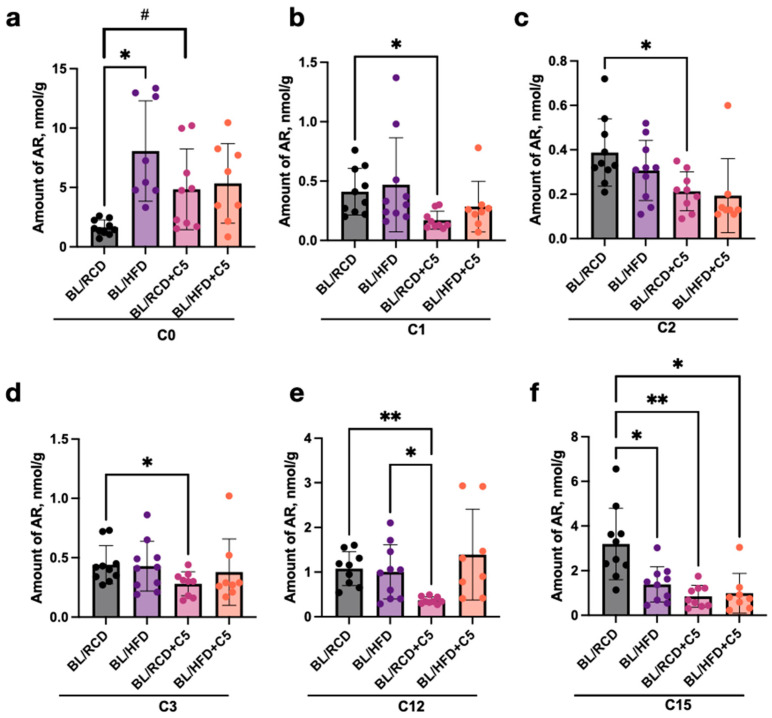
Differences in AR content (nmol/g) in stool samples of C57BL/6 mice after feeding with regular (BL/RCD) or high-fat (BL/HFD) diet without or with C5 (sign “+C5”) supplementation: (**a**) C0, resorcinol, (**b**) C1, methylresorcinol, (**c**) C2, ethylresorcinol, (**d**) C3, propylresorcinol, (**e**) C12, dodecylresorcinol, and (**f**) C15, pentadecylresorcinol. Comparison of AR profiles was performed using the one-way Welch’s ANOVA test followed by the *t* test (* *p* < 0.05, ** *p* < 0.01 for ANOVA and # *p* < 0.05 for the *t* test).

**Figure 4 ijms-24-14206-f004:**
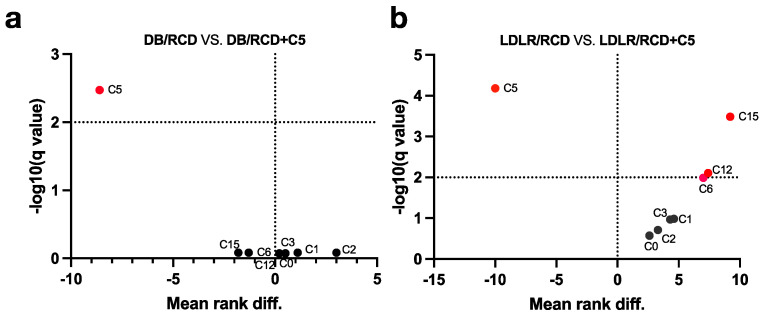
Volcano plots based on multiple Mann–Whitney tests of (**a**) *db/db* mice (DB) or (**b**) LDLR (−/−) mice (LDLR) fed a regular chow diet (RCD) with or without olivetol (+C5) supplementation. The volcano graph represents the changes in AR levels between different dietary conditions. The Q value reflects a false discovery rate of 5%. The mean rank difference values reflect the direction of changes in the AR level (values below zero indicate an increase in the AR level, while values above zero indicate a decrease in the AR level in the faeces of mice fed a diet with C5 supplementation). Statistically significant values are marked with red dots.

**Figure 5 ijms-24-14206-f005:**
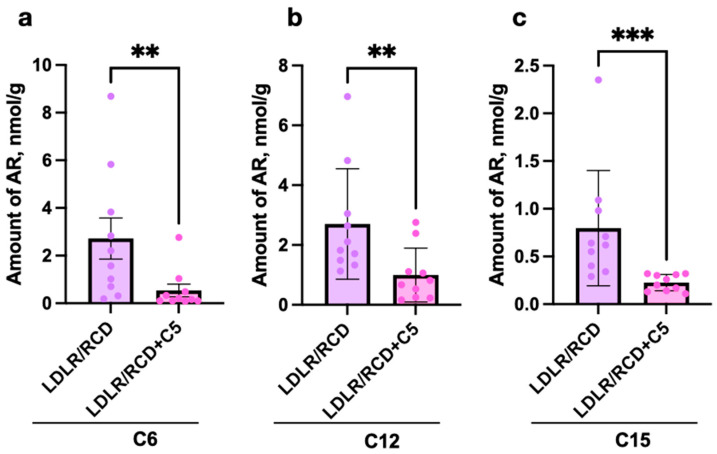
Differences in AR content in stool samples from LDLR (−/−) mice after feeding with RCD without or with C5 (sign ‘+C5’) supplementation according to unpaired *t* tests: (**a**) C6, hexylresorcinol, (**b**) C12, dodecylresorcinol, and (**c**) C15, pentadecylresorcinol (** *p* < 0.01, *** *p* < 0.001).

**Figure 6 ijms-24-14206-f006:**
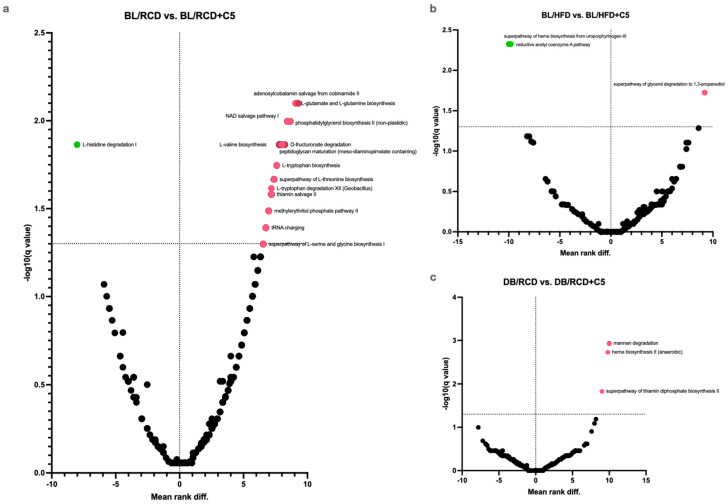
Volcano plots showing differences in the abundance of metabolic pathways of the gut microbiota based on multiple Mann–Whitney tests of C57BL/6 (BL) (**a**,**b**) or db/db (DB) (**c**) mice fed a regular chow diet (RCD) (**a**,**c**) or a high fat diet (HFD) (**b**) with (+C5) or without olivetol supplementation. The volcano plot represents changes in the abundance of metabolic pathways between different dietary conditions. The Q value reflects a false discovery rate of 5%. The mean rank difference values reflect the direction of changes in the abundance of metabolic pathways (values below zero indicate an increased representation of the pathways, while values above zero indicate a decreased representation of the pathways in the microbiota of mice fed a diet with C5 supplementation). Statistically significant values are labelled with green and red dots.

**Figure 7 ijms-24-14206-f007:**
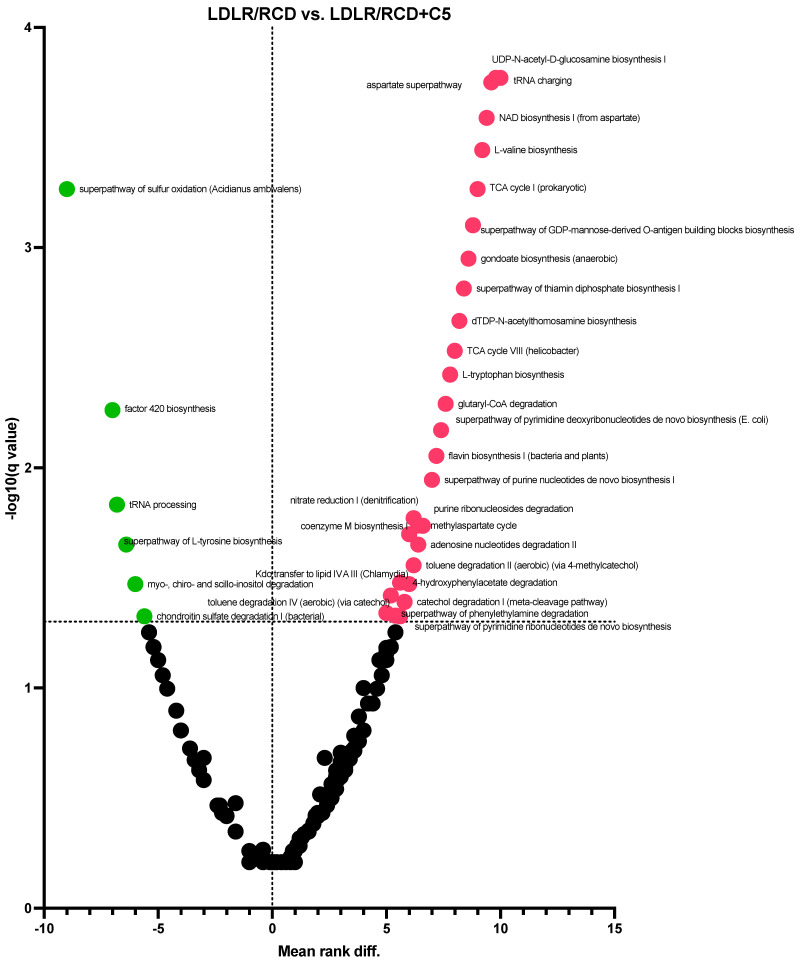
Volcano graph showing differences in the abundance of metabolic pathways of the gut microbiota based on multiple Mann–Whitney tests of LDLR (−/−) mice fed a regular diet of food (RCD) with (+C5) or without olivetol supplementation. The volcano plot represents changes in the abundance of metabolic pathways between different dietary conditions. The Q value reflects a false discovery rate of 5%. The mean rank difference values reflect the direction of the changes in the abundance of metabolic pathways (values below zero indicate a higher representation of the pathway, while values above zero indicate a decreased representation of the pathway in the microbiota of mice fed a diet with C5 supplementation). Statistically significant values are marked with green or red dots.

**Table 1 ijms-24-14206-t001:** Comparative analysis of several metabolic pathway abundance in mice fed a regular chow diet (RCD) or high-fat diet (HFD) in mice with different genetic background. Arrows show changes in relative pathway abundance (increased abundance is shown by the arrow directed up and decreased abundance is shown by the arrow directed down) after C5 supplementation or in HFD compared to RCD in C57/BL6SPF mice.

	Superpathway of Sulfur Oxidation (Acidianus Ambivalens)	Reductive Acetyl Coenzyme A Pathway	Factor 420 Biosynthesis	Superpathway of Haem Biosynthesis from Uroporphyrinogen-III/Haem Biosynthesis II (Anaerobic)	Catechol Degradation	Thiamine Biosynthesis/Salvage	Methanogenesis from Acetate	Adenosylcobalamine Synthesis/Salvage	L-Histidine Degradation I	Mannan Degradation	Synthesis of Fatty Acids (Palmitate, Oleate, Stearate, etc.)
BL/RCD vs. BL/HFD										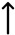	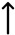
BL/RCD vs. BL/RCD + C5									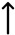		
BL/HFD vs. BL/HFD + C5		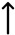		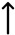							
DB/RCD vs. DB/RCD + C5											
LDLR/RCD vs. LDLR/RCD + C5	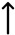		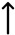								

**Table 2 ijms-24-14206-t002:** Characteristics of groups of germ-free mice and donors of stool samples for faecal microbiota transplantation. N/A—not applicable, yo—years old.

Group of Animals	Number of Animals	Material for Transplantation	Route of Administration	Microbiota Donor
1 (control)	10	0.9% NaCl solution	Intragastric	N/A
2	10	Faecal microbiota sample from Donor 1	Intragastric	Donor 1. Male, 42 yo
3	10	Faecal microbiota sample from Donor 2	Intragastric	Donor 2. Male, 36 yo
4	10	Faecal microbiota sample from Donor 3	Intragastric	Donor 3. Female, 28 yo

## Data Availability

The data presented in this study are available on request from the corresponding author. The data are not publicly available due to unfinished patenting process.

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
