# Peer review of "Alkylresorcinols as New Modulators of the Metabolic Activity of the Gut Microbiota"

_ijms, 2023, doi:10.3390/ijms241814206_

Round 1
Reviewer 1 Report
I have reviewed the manuscript titled "Alkylresorcinols of the human gut microbiota: a new way to influence holobiont’s functioning " and appreciate the authors' efforts in tackling an interesting topic. I commend the authors for their enthusiasm in exploring this topic. The topic holds promise for contributing valuable insights to the field. However, I have several concerns regarding the scientific soundness, clarity of aims, experimental design, figures, statistics, and conclusions presented in the article.
This article lacks a clear articulation of its research objectives and hypotheses, making it challenging to understand the significance of the study. The research aims should be explicitly stated to provide a framework for the entire investigation. Without this foundation, the article appears to lack a coherent direction.
The experimental design is notably flawed. Critical details about the methodology, controls, and variables are missing, rendering the validity and reproducibility of the study questionable. A more robust and well-defined experimental approach is necessary to draw meaningful conclusions.
The figures included in the manuscript are unclear and inadequately labeled. Proper labeling and clear, high-quality visuals are essential for conveying the experimental results effectively. Additionally, the statistical analysis employed raises concerns. The methods used should be thoroughly explained and justified. It's crucial to provide a clear rationale for the chosen statistical techniques and their appropriateness for the data.
The conclusions drawn in the article seem to exceed the scope of the evidence presented. The authors should refrain from overstating the implications of their findings. The conclusions should be revised to accurately reflect the limitations of the study and the extent to which the results can support the claims made.
Suggestions:
1. Clearly define the research objectives and hypothesis at the outset of the article.
2. Revise the experimental design section to provide comprehensive details about methodology, controls and variables.
3. Enhance the clarity and quality of figures by improving labeling and visual representation.
4. Provide a detailed explanation for the statistical methods used and their appropriateness for the data.
5. Revise the conclusions with the actual evidence presented and acknowledge the study's limitations.
English needs moderate editing.
Author Response
Main comment. Dear Reviewers, we appreciate your attentive analysis of our research and your censorious remarks that will allow us to improve our study. Based on the comments of the reviewers, we significantly revised our manuscript. We also added some additional data obtained from bioinformatic analysis that we have performed previously to describe metabolic alterations that occur in the mouse gut under conditions of olivetol supplementation. We must note that high-throughput sequencing analysis of the mouse gut microbiota metagenome was performed as part of the experimental design. Additionally, we performed a bioinformatic analysis of the metabolic pathways of the microbiota and the representation of enzymes. However, we cannot publish the results of taxonomy analysis, alpha-diversity, and biochemical features of mice investigated because we are on a way of patenting some issues of the research. Thus, we cannot disclose this information until the patent process is completed. That is why here in the manuscript we can only provide the data of metabolic pathway reconstruction, although we can provide all raw data as well as statistic data on the request of the editors or reviewers. Here, you can see our responses to the reviewers’ comments.
Reviewer 1.
- “I have reviewed the manuscript titled "Alkylresorcinols of the human gut microbiota: a new way to influence holobiont’s functioning " and appreciate the authors' efforts in tackling an interesting topic. I commend the authors for their enthusiasm in exploring this topic. The topic holds promise for contributing valuable insights to the field. However, I have several concerns regarding the scientific soundness, clarity of aims, experimental design, figures, statistics, and conclusions presented in the article. This article lacks a clear articulation of its research objectives and hypotheses, making it challenging to understand the significance of the study. The research aims should be explicitly stated to provide a framework for the entire investigation. Without this foundation, the article appears to lack a coherent direction.”
We agree with the comments of the referee and have tried to improve the clearness, scientific soundness, figures, statistics, and conclusions of the research. As you can see in revised version of the manuscript, the abstract, aims, results, discussion, and conclusion sections were changed.
- “The experimental design is notably flawed. Critical details about the methodology, controls, and variables are missing, rendering the validity and reproducibility of the study questionable. A more robust and well-defined experimental approach is necessary to draw meaningful conclusions.”
See the main comment concerning the design of the experiment.
- “The figures included in the manuscript are unclear and inadequately labeled. Proper labeling and clear, high-quality visuals are essential for conveying the experimental results effectively.”
All the figures were corrected and re-labelled. Additional figures have been added.
- “Additionally, the statistical analysis employed raises concerns. The methods used should be thoroughly explained and justified. It's crucial to provide a clear rationale for the chosen statistical techniques and their appropriateness for the data.”
Statistical analysis was revised. All data were recalculated. We provide a description of the methods used to evaluate statistically significant results.
- “The conclusions drawn in the article seem to exceed the scope of the evidence presented. The authors should refrain from overstating the implications of their findings. The conclusions should be revised to accurately reflect the limitations of the study and the extent to which the results can support the claims made.”
We agree. The conclusions were completely revised.
- “Comments on the Quality of English Language English needs moderate editing.”
We edited the manuscript and tried to eliminate all linguistic errors.
Reviewer 2 Report
Dear authors,
I believe that the role of polyphenolic compounds on gut physiology is a very interesting area because these kind of products are related with human and animal health by its effects on different gut populations of bacteria. Your work presented for publication in the International Journal of Molecular Sciences is centred in the study of alkylresorcinols (ARs), a subgroup of polyphenolic compounds produced by different bacterial populations of human gut, which can regulate the intestinal microbiota composition and the host metabolism.
Nevertheless, I consider that your document has some minor comments and, unfortunately, some other major comments that need your consideration before its inclusion in the International Journal of Molecular Sciences.
Major comments:
I believe that some additional data that are not included in the present document will be useful to understand what happens in the gut microbiota and in the intestinal physiology in relationship with the synthesis of ARs by different bacterial populations and, reciprocally, the changes of bacterial populations derived by the production or administration of ARs in your experimental model.
I consider that a second set of faecal samples to analyse the gut microbiota composition of experimental animals at different times would have been very useful to know which bacteria populations are involved in the synthesis of different ARs, and which bacteria populations change in the presence of different ARs. After the revision of your previous article published in the Biology Bulletin (2022) to know the changes observed on gut microbiota in the different experimental groups, I observe that your results were centred on the data of diversity and on the changes of microbiota at very high phylogenetical level (phylum, class or order), but without information of microbiota at family, genus or species level, the phylogenetical levels more closely related with the production or not of different secondary metabolites. This information can be obtained using the primers 337F and 805R that you used in your microbiota analysis.
Additionally, the study of gut microbiota using 16S RNA sequencing produce information that can be processed using the PICRUSt package, a bioinformatics software designed to predict metagenome functional content from marker gene as 16S RNA is. PICRUSt software permits to identify which OTUs are contributing with defined functions, and can collapse different predicted functions into higher categories (e.g. pathway of ARs synthesis).
Because the effect of ARs on intestinal stability alters the permeability of lipopolysaccharide (LPS) to blood stream, as you remarked in your manuscript, the measure of plasmatic LPS in the experimental animals would have been a very interesting additional data that could correlate the changes in gut microbiota with the production of ARs and the intestinal physiology. The plasmatic LPS is directly associated with the increase of intestinal permeability and can be linked with different functions related with the intestinal mucosa integrity and its barrier effect. Additionally, the increase of the absorption of LPS is related with the production of proinflammatory cytokines and a reduction of health status of humans and animals.
Another major comment to your document is related with the results associated to Figure 4.b. If you maintain the same symbology than in the other figures, the absence of asterisks means the lack of statistically significant differences between experimental groups, and because of that you can’t incorporate the phrases included in lines 181 and 182.
Minor comments:
Line 58: Please, change “Bacteria” by “bacteria”.
Lines 89-90: please, change “B. cereus” by “Bacillus cereus”.
Line 104-105: Please, change "Extensive studying of eCB system functioning” by “The extensive study of eCB system functions”.
Figure 1: Please remark the meaning of asterisk symbol in statistical analysis results (* means p value ≤ 0.05, ** means p value ≤ 0.01, *** means p value ≤ 0.001.
Line 145: Please, change “to investigation of” by “to investigate the”.
Line 184: Please, after “RCD or RCD+C5 nutrition” include “(data not shown)”.
Lines 208-210: Please, change the sentence, because no statistically significant differences were obtained, as previously I have mentioned in the major comments section.
Lines 221-222: The gut microbiota deep sequencing and PICRUSt analysis could help to clarify this point.
Lines 234-236: The plasmatic LPS analysis, together with the gut microbiota deep sequencing of experimental animals, could help to clarify this point.
See the comments and suggestions for authors.
Author Response
Main comment. Dear Reviewers, we appreciate your attentive analysis of our research and your censorious remarks that will allow us to improve our study. Based on the comments of the reviewers, we significantly revised our manuscript. We also added some additional data obtained from bioinformatic analysis that we have performed previously to describe metabolic alterations that occur in the mouse gut under conditions of olivetol supplementation. We must note that high-throughput sequencing analysis of the mouse gut microbiota metagenome was performed as part of the experimental design. Additionally, we performed a bioinformatic analysis of the metabolic pathways of the microbiota and the representation of enzymes. However, we cannot publish the results of taxonomy analysis, alpha-diversity, and biochemical features of mice investigated because we are on a way of patenting some issues of the research. Thus, we cannot disclose this information until the patent process is completed. That is why here in the manuscript we can only provide the data of metabolic pathway reconstruction, although we can provide all raw data as well as statistic data on the request of the editors or reviewers. Here, you can see our responses to the reviewers’ comments.
- “..Major comments:
I believe that some additional data that are not included in the present document will be useful to understand what happens in the gut microbiota and in the intestinal physiology in relationship with the synthesis of ARs by different bacterial populations and, reciprocally, the changes of bacterial populations derived by the production or administration of ARs in your experimental model.
I consider that a second set of faecal samples to analyse the gut microbiota composition of experimental animals at different times would have been very useful to know which bacteria populations are involved in the synthesis of different ARs, and which bacteria populations change in the presence of different ARs. After the revision of your previous article published in the Biology Bulletin (2022) to know the changes observed on gut microbiota in the different experimental groups, I observe that your results were centred on the data of diversity and on the changes of microbiota at very high phylogenetical level (phylum, class or order), but without information of microbiota at family, genus or species level, the phylogenetical levels more closely related with the production or not of different secondary metabolites. This information can be obtained using the primers 337F and 805R that you used in your microbiota analysis.
Additionally, the study of gut microbiota using 16S RNA sequencing produce information that can be processed using the PICRUSt package, a bioinformatics software designed to predict metagenome functional content from marker gene as 16S RNA is. PICRUSt software permits to identify which OTUs are contributing with defined functions, and can collapse different predicted functions into higher categories (e.g. pathway of ARs synthesis).”
We totally agree with the comments of the Reviewer. We add information concerning PICRUSt analysis, however, we cannot provide taxonomic analysis in present study (see the main comment above).
- “Because the effect of ARs on intestinal stability alters the permeability of lipopolysaccharide (LPS) to blood stream, as you remarked in your manuscript, the measure of plasmatic LPS in the experimental animals would have been a very interesting additional data that could correlate the changes in gut microbiota with the production of ARs and the intestinal physiology. The plasmatic LPS is directly associated with the increase of intestinal permeability and can be linked with different functions related with the intestinal mucosa integrity and its barrier effect. Additionally, the increase of the absorption of LPS is related with the production of proinflammatory cytokines and a reduction of health status of humans and animals.”
We have decided to put away the data regarding the association with LPS concentration to avoid potential copyright issues. But we agree that plasmatic LPS measurement will help to track physiological changes in mice's gut. This point is planned in the next research.
- “Another major comment to your document is related with the results associated to Figure 4.b. If you maintain the same symbology than in the other figures, the absence of asterisks means the lack of statistically significant differences between experimental groups, and because of that you can’t incorporate the phrases included in lines 181 and 182.”
All the figures were corrected and re-labeled. Additional figures have been added. Statistical analysis was revised. All data were recalculated. We provide a description of the methods used to evaluate statistically significant results.
Minor comments:
Line 58: Please, change “Bacteria” by “bacteria”.
Ö Done
Lines 89-90: please, change “B. cereus” by “Bacillus cereus”.
Ö Done
Line 104-105: Please, change "Extensive studying of eCB system functioning” by “The extensive study of eCB system functions”.
Ö Done
Figure 1: Please remark the meaning of asterisk symbol in statistical analysis results (* means p value ≤ 0.05, ** means p value ≤ 0.01, *** means p value ≤ 0.001.
Revised
Line 145: Please, change “to investigation of” by “to investigate the”.
Ö Done
Line 184: Please, after “RCD or RCD+C5 nutrition” include “(data not shown)”.
Revised
Lines 208-210: Please, change the sentence, because no statistically significant differences were obtained, as previously I have mentioned in the major comments section.
Ö Done
Lines 221-222: The gut microbiota deep sequencing and PICRUSt analysis could help to clarify this point.
See the comments above.
Lines 234-236: The plasmatic LPS analysis, together with the gut microbiota deep sequencing of experimental animals, could help to clarify this point.
See the comments above.
Round 2
Reviewer 1 Report
Dear Authors,
The revised manuscript has improved compared to previous version. However, there needs to be significant modification of the manuscript to prove the claim that gut microbiota is producing alkyl resorcinol.
1. Figure 1 is showing the quantitative difference of AR between control (non-FMT transplanted) and three tests (FMT transplanted from 3 donors) on day 14 only. However, this single time point does not provide conclusive evidence to support the assertion put forth in the paper.
To enhance the comprehensiveness of the study, it is very important to present a time-course representation that demonstrates the progressive increase in ARs across all sets of mice, commencing from day 0.
2. Figure 2: The bottom panel mentioned as GC-MS should be presented with additional details. Please increase the resolution of the entire figure while ensuring clear labeling and comprehensive explanations.
3. Fig 2: Please provide insights into the rationale behind selecting these specific mice genotypes for the study.
4. Figure 3: There is minimal explanation given on how the experiment was conducted. Elaborate the Methods section.
How long were the mice fed with the specific diet? A longitudinal study tracking alkyl resorcinol levels over time could provide a more comprehensive understanding of the effects of different diets.
What is indicated by “#” showing statistical significance between BL/RCD and BL/RCD +C5?
5. Fig 3: Please consider providing further elaboration of the discussion section. It is recommended to provide deeper insights into the experimental design, the context of the dietary choices and the implications of the findings within the broader scope of the dietary and microbiota influences on AR production by bacteria.
Please discuss the rationale behind studying the impact of a regular chow diet Vs high fat diet on AR production. Contrasting the effects of a gluten-free diet Vs gluten-containing diet on AR concentrations would have provided valuable insights in this regard.
6. Fig 4: In the volcano plot, it is labelled DB/RCD Vs DB/RCD+C5. Please explain from which experiment this data is generated. In the context of your study, the volcano plot could represent the changes in the alkyl resorcinol levels between different dietary conditions, providing a visual summary of the magnitude of changes and their associated statistical significance. Properly labeling the axes and providing a clear legend would enhance the interpretability of the plot for the readers.
7. Providing data from 16S rRNA sequencing is very essential in a study that claims that microbiota is involved in AR production. Such data would enable a comprehensive understanding of the microbial community changes that might have contributed to the observed AR production. It is important for the overall credibility of the study.
8. Fig 6: Label the volcano plot in a standard manner.
9. Discussion part must be significantly improved. Even though the specific pathways in bacteria that are directly involved in the AR production have not been extensively characterized, it is important to highlight certain metabolic pathways in the discussion part that could be significantly playing a role based on your data.
For example, ARs share structural similarities with phenolic compounds, which can be derived from aromatic amino acids. Enzymes involved in aromatic amino acid metabolism might contribute to AR synthesis. You could discuss on that part.
10. “The ability of C5 to act as CB1/CB2 ligands suggests the connection of AR regulatory activity and the functioning of the eCB system.” This study has not addressed this point. It is not necessary to mention this here.
11. It is an overstatement to say that based on this study it can be concluded that ARs may serve as QS molecules influencing gut microbiota composition and host metabolism.
12. By presenting a more balanced assessment of the study’s implications and acknowledging the need for further research, a revised conclusion will maintain a reasonable and truthful perspective on the significance of your findings.
Editing of English language is needed.
Author Response
Dear Reviewer! Thank you very much for attentive consideration of our investigation and the important issues that you raise here for discussion.
“1. Figure 1 is showing the quantitative difference of AR between control (non-FMT transplanted) and three tests (FMT transplanted from 3 donors) on day 14 only. However, this single time point does not provide conclusive evidence to support the assertion put forth in the paper.
To enhance the comprehensiveness of the study, it is very important to present a time-course representation that demonstrates the progressive increase in ARs across all sets of mice, commencing from day 0.”
According to the design of the experiment, we kept mice on a standard diet (regular chow) for 14 days after fecal microbiota transplantation. We must note that the diet did not change before, during and after transplantation. The 14-day period is sufficient to avoid environmental AR contamination and induce AR synthesis by transplanted microbes. Furthermore, in all experiments where we were analyzing fecal compounds or the microbiota, we took a sample of feces directly from the colon of each of the 10 animals in each group. We believe that the use of a large number of animals to provide more time points will be inhumane. That is why we cannot provide more points to present a time-course representation that demonstrates the progressive increase in ARs across all sets of mice. However, we agree that this single time point does not provide conclusive evidence to support the claim made in the paper and therefore we write “a significant increase in the content of C3, C12 and C15 (p <0.01) in feces indicates a high probability of synthesis of these ARs by representatives of the human gut microbiota transplanted into gnotobiotic mice” (see lines 138-140). We believe that sufficient evidence would be the sowing microbes able to synthesize AR in culture medium and the identification of such microbes in the case these microbes are cultured. Without a doubt, a time-course evaluation of AR content is necessary but impossible under the condition of our experiment. The limitations of the study were discussed in the Conclusions part.
“2. Figure 2: The bottom panel mentioned as GC-MS should be presented with additional details. Please increase the resolution of the entire figure while ensuring clear labeling and comprehensive explanations.”
We changed the figure 2 and also provided mass-spectrums of different ARs, investigated in our study (see Supplementary Figures 1-9).
“3. Fig 2: Please provide insights into the rationale behind selecting these specific mice genotypes for the study.”
AR are known to have beneficial effects on health. For example, chronic supplementation with wheat AR prevents obesity and its associated metabolic symptoms induced by a high-fat high-sucrose diet in mice (https://dx.doi.org/10.3945/jn.114.202754). ARs increase glucose tolerance and insulin sensitivity by suppressing hepatic lipid accumulation and intestinal cholesterol absorption, which subsequently suppresses diet-induced obesity in mice. On the other hand, ARs improve feed efficiency by decreasing dietary lipid absorption (https://dx.doi.org/10.1016/j.nut.2022.111796). Therefore, ARs are involved in lipid metabolism and could have protective properties in obesity. In this investigation, our aim was to study the effects of AR on intestinal microbiota function in relation to diet-induced or genetically determined obesity. Db/db mice are known to develop an obese phenotype independently of diet type, whereas ldlr(-/-) mice on the contrary do not develop an obese phenotype under conditions of a regular diet but have abnormal lipid metabolism leading to the manifestation of atherosclerosis. Different studies have confirmed that obese and lean mice have a different composition and metabolic activity of the gut microbiota. We proposed that microbiota metabolic activity is associated not only with phenotype but also with mouse genotype. Therefore, AR could influence the metabolic activity of the gut microbiota differently in genetically determined or diet-induced obesity. As we can see, the metabolic profile of the mouse microbiota differs significantly after olivetol supplementation. In particular, C5 supplementation inverses the abundance of several pathways that were different in mice fed a HFD compared to mice fed an RCD (see the table below). For instance, the abundance of fatty acid synthesis and mannan degradation pathways increased in mice fed a high-fat diet compared to mice fed a regular chow diet, while the abundance of the reductive acetyl coenzyme A pathway and the heme biosynthesis pathway decreased. Supplementation of C5 to a HFD increases the abundance of heme biosynthesis and reductive acetyl coenzyme A pathways, supplementation of C5 to db/db mice (who develop obesity as mice fed a HFD do) decreases the mannan degradation pathway, supplementation of C5 to ldlr(-/) mice (who do not develop obesity) increases pathways of methanogenesis (s/p of sulphur oxidation and p/w of factor 420 biosynthesis), which were decreased in mice fed an HFD. These may indicate the ability of C5 to influence the metabolic activity of the microbiota. We can suggest that AR due to its ability to decrease the absorption of dietary lipids changes the representation of the substate to gut microbes, thus promoting bacteria to adapt to a new 'fodder base'. However, depending on the specific genetic background, such adaptations may have opposite directions. As we can see in db/db and ldlr(-/-) mice, C5 supplementation decreases the abundance of pathways involved in methanogenesis in db/bd mice and increases in ldlr(-/-) mice.
At present, our knowledge concerning the exact mechanisms of AR action is limited. However, we see a great regulatory potential for AR in modulating gut microbiota composition and/or its metabolic activity. Without a doubt, this area of investigation requires meticulous research.
|
Superpathway of sulfur oxidation (Acidianus ambivalens)
|
Reductive acetyl coenzyme A pathway
|
Factor 420 biosynthesis
|
Superpathway of heme biosynthesis from uroporphyrinogen-III/ heme biosynthesis II (anaerobic)
|
Catechol degradation |
Thiamine biosynthesis/ salvage |
Methanogenesis from acetate |
Adenosylcobalamine synthesis/salvage |
L-histidine degradation I |
Mannan degradation |
Synthesis of fatty acids (palmitate, oleate, stearate, etc.) |
BL/RCD vs. BL/HFD |
↓ |
↓ |
|
↓ |
↓ |
↓ |
↓ |
↓ |
↓ |
↑ |
↑ |
BL/RCD vs. BL/RCD+C5 |
|
|
|
↓ |
|
↓ |
↓ |
↓ |
↑ |
|
|
BL/HFD vs. BL/HFD+C5 |
|
↑ |
|
↑ |
|
|
|
|
|
|
|
DB/RCD vs. DB/RCD+C5 |
↓ |
|
|
↓ |
|
↓ |
|
|
|
↓ |
|
LDLR/RCD vs. LDLR/RCD+C5 |
↑ |
↓ |
↑ |
↓ |
↓ |
↓ |
↓ |
|
↓ |
|
↓ |
“4. Figure 3: There is minimal explanation given on how the experiment was conducted. Elaborate the Methods section.
How long were the mice fed with the specific diet? A longitudinal study tracking alkyl resorcinol levels over time could provide a more comprehensive understanding of the effects of different diets.
What is indicated by “#” showing statistical significance between BL/RCD and BL/RCD +C5?”
The Methods section was revised.
Comparison of AR profiles was performed using the one-way Welch’s ANOVA test followed by the t test (*p < 0.05, **p < 0.01 for ANOVA and #p < 0.05 for the t test). The figure legend was corrected.
“5. Fig 3: Please consider providing further elaboration of the discussion section. It is recommended to provide deeper insights into the experimental design, the context of the dietary choices and the implications of the findings within the broader scope of the dietary and microbiota influences on AR production by bacteria.
Please discuss the rationale behind studying the impact of a regular chow diet Vs high fat diet on AR production. Contrasting the effects of a gluten-free diet Vs gluten-containing diet on AR concentrations would have provided valuable insights in this regard. “
As we mentioned above, ARs are involved in lipid metabolism and could have protective properties against obesity. ARs are also used as markers of gluten-containing or cereal products intake due to their high content in cereal bran. However, cereals mainly contain long alkyl chain resorcinols (C17-C25), and the effects of whole grain dietary products are conditioned by these ARs (https://www.ncbi.nlm.nih.gov/pmc/articles/PMC8640985/). In the case of our study, we did not want to investigate the effects of cereal AR on the production of ARs by the intestinal microbiota, but rather we wanted to understand 1) is there any influence of olivetol, which is known to be a ligand for endocannabinoid receptors, on the composition of the intestinal microbiota and its metabolic activity; 2) is there an interrelationship between different ARs (potentially synthesized in the intestinal tract and received from food). These influences may be crucial for the formation of a specific 'obese' microbiota and the development of the obese phenotype. Therefore, we chose a high-fat diet and Lepr-null mice for modelling the obese phenotype and ldlr-null mice to model abnormalities of lipid metabolism.
“6. Fig 4: In the volcano plot, it is labelled DB/RCD Vs DB/RCD+C5. Please explain from which experiment this data is generated. In the context of your study, the volcano plot could represent the changes in the alkyl resorcinol levels between different dietary conditions, providing a visual summary of the magnitude of changes and their associated statistical significance. Properly labeling the axes and providing a clear legend would enhance the interpretability of the plot for the readers.”
We corrected the legends.
- Providing data from 16S rRNA sequencing is very essential in a study that claims that microbiota is involved in AR production. Such data would enable a comprehensive understanding of the microbial community changes that might have contributed to the observed AR production. It is important for the overall credibility of the study.
We provide data from 16S rRNA sequencing, as well as the list of annotated OTUs, to the editor. However, as we mentioned earlier, we cannot present these data in publication because we are patenting a symbiotic product, and according to the nondisclosure agreement, we are not allowed to publish microbiota taxonomy analysis data. But we are thinking of publishing all sequencing data later (after patent acceptation) providing a comparative study of the gut microbiota composition of mice fed a different diet according to the genetic background.
Fig 6: Label the volcano plot in a standard manner.
The graph was corrected.
9. Discussion part must be significantly improved. Even though the specific pathways in bacteria that are directly involved in the AR production have not been extensively characterized, it is important to highlight certain metabolic pathways in the discussion part that could be significantly playing a role based on your data.
For example, ARs share structural similarities with phenolic compounds, which can be derived from aromatic amino acids. Enzymes involved in aromatic amino acid metabolism might contribute to AR synthesis. You could discuss on that part.
As we know from publications enzymes involved in aromatic amino acid metabolism are not implicated into AR synthesis. Thus, it looks too speculative and we afraid to draw such conclusions. Nevertheless, the Discussion part was revised. See the manuscript.
10. “The ability of C5 to act as CB1/CB2 ligands suggests the connection of AR regulatory activity and the functioning of the eCB system.” This study has not addressed this point. It is not necessary to mention this here.
We agree and have corrected the manuscript.
11. It is an overstatement to say that based on this study it can be concluded that ARs may serve as QS molecules influencing gut microbiota composition and host metabolism.
Corrected.
12. By presenting a more balanced assessment of the study’s implications and acknowledging the need for further research, a revised conclusion will maintain a reasonable and truthful perspective on the significance of your findings.
The limitations of the study were added to the Conclusions part.
Reviewer 2 Report
Thanks for your very extensive modifications of the original document in accordance with my preliminary indications. I consider that the inclusion of the metabolomic study of faecal microbiota conducted by PICRUSt permits to infer significant changes in microbiota composition between the experimental groups, but I think that information about microbial changes is an important part of your work, because one of the principal roles of alkylresorcinols is the effect on different gut populations of bacteria. Nevertheless, as you mentioned in the “Main comment” paragraph included in your responses to my first evaluation, these data have not been included by the probable incompatibility with possible patents (symbiotics?) that are in process. Because of that I have asked a question to IJMS’s editors to know if this information occultation is sufficiently justified.
If this information occultation is considered no problematic by the editors, I believe that the last version of your document can be included in the International Journal of Molecular Sciences with only an additional minor modification in lines 80 and 81, related with the type of letter used for the mentioned phyla. At present, only bacterial taxonomy level of genus and species are written in italic type letter.
Author Response
“Thanks for your very extensive modifications of the original document in accordance with my preliminary indications. I consider that the inclusion of the metabolomic study of faecal microbiota conducted by PICRUSt permits to infer significant changes in microbiota composition between the experimental groups, but I think that information about microbial changes is an important part of your work, because one of the principal roles of alkylresorcinols is the effect on different gut populations of bacteria. Nevertheless, as you mentioned in the “Main comment” paragraph included in your responses to my first evaluation, these data have not been included by the probable incompatibility with possible patents (symbiotics?) that are in process. Because of that I have asked a question to IJMS’s editors to know if this information occultation is sufficiently justified.
If this information occultation is considered no problematic by the editors, I believe that the last version of your document can be included in the International Journal of Molecular Sciences with only an additional minor modification in lines 80 and 81, related with the type of letter used for the mentioned phyla. At present, only bacterial taxonomy level of genus and species are written in italic type letter.”
Dear Reviewer! Thank you very much for attentive consideration of our investigation and high evaluation of our study.
The manuscript was revised. The names of phyla have been corrected.
We provide data from 16S rRNA sequencing, as well as the list of annotated OTUs, to the editor. However, as we mentioned earlier, we cannot present these data in publication because we are patenting a symbiotic product, and according to the nondisclosure agreement, we are not allowed to publish microbiota taxonomy analysis data. But we are thinking of publishing all sequencing data later (after patent acceptation) providing a comparative study of the gut microbiota composition of mice fed a different diet according to the genetic background.
Round 3
Reviewer 1 Report
Dear Authors,
Thank you for your responses and the modifications.
Further comments.
1. I think the GC-MS data presentation could be improved for publication.
I would suggest to look at the GC-MS data presentation in the following research article as an example. Also, please label the x and y axis.
"Determination of alkylresorcinols and their metabolites in biological samples by gas chromatography–mass spectrometry."Wierzbicka R, Wu H, Franek M, Kamal-Eldin A, Landberg R.
2. Please label all the figures in the article properly.
For example, there are three images in Figure 1 is actually Fig 1A, Fig 1B, Fig 1 C.
Please label each figure as A, B, C and mention the label correctly in the figure legend as well as in the main text.
Figure 6 has a,b and c labels, but it is not mentioned in the figure legend. Please maintain consistency throughout the article in the way figures are labelled and described.
Thank you.
Please go through the article to verify English grammar.
One example, line 264 is not grammatically correct.
"Furthermore, olivetol serves as a negative regulator of ARs C3, C12 and C15, which we have shown may have the origin of the gut microbiota.
Author Response
Dear reviewer, thank you for attentive consideration of our manuscript.
- I think the GC-MS data presentation could be improved for publication.
The presentation of GC-MS data was modified according to the recommendations. See Supplementary Figures 1-9.
- Please label all the figures in the article properly.
The figure labels were corrected. See the manuscript.
- Please go through the article to verify English grammar.
We corrected the language.